# Exploring infection prevention and control knowledge and beliefs in the Solomon Islands using Photovoice

**Vanessa L. Sparke**[1]*, **David MacLaren**[2]☺, **Dorothy Esau**[3]☺, **Caryn West**[1]☺

**1** College of Healthcare Sciences, James Cook University, Townsville, Queensland, Australia, **2** College of Medicine and Dentistry, James Cook University, Townsville, Queensland, Australia, **3** Baru Conservation Alliance, East Kwaio, Solomon Islands

☺ These authors contributed equally to this work.
* vanessa.sparke1@jcu.edu.au

**Data Availability Statement:** Data are provided within the manuscript in the form of verbatim statements from participants, and are publicly

## Abstract

Healthcare associated infections are the most common complication of a person's hospital stay. Contemporary infection prevention and control programs are universally endorsed to prevent healthcare associated infections. However, western biomedical science on which contemporary infection prevention and control is based, is not the only way that staff and patients within healthcare settings understand disease causation and/or disease transmission. This results paper reports on one aspect of a study which ascertains perceptions of disease transmission and how these influence infection prevention and control practice at Atoifi Adventist Hospital Solomon Islands. Photovoice was used as the primary data collection method with staff and patients. The germ theory and hospital hygiene processes were only one of many explanations of disease transmission at the hospital. Many social, cultural and spiritual influences played an important role in how people understood disease to be transmitted. Although infection prevention and control models based on western science continue to form the premise of reducing healthcare associated infections in Solomon Islands and locations across the globe, local social, cultural and spiritual beliefs need to be considered when planning and implementing infection prevention and control programs to ensure success.

## Introduction

Infection Prevention and Control (IP&C) programs in healthcare settings aim to reduce disease transmission between patients, visitors and staff. Healthcare associated infections (HAI) are the most common complication of a person's hospital stay and are an ongoing cause of morbidity, mortality and excess healthcare expenditure [1,2]. Contemporary biomedical (Western) based IP&C programs were only formalised in the 1980s as a result of the Study on the Effectiveness of Nosocomial Infection Control (SENIC) [3], which proved that comprehensive infection control programs can reduce healthcare associated infection rates. Before these formal contemporary IP&C programs, the deliberate intent to reduce infectious agents

available from the James Cook University repository (https://doi.org/10.25903/4tqh-mt24).

**Funding:** The authors received no specific funding for this work.

**Competing interests:** The authors have declared that no competing interests exist.

passing between people in public health and hospital settings had only been occurring for little more than a century [3]. It was only in the 1850's that Holmes and Semmelweis, separately published findings on the transmission of puerperal sepsis and described the role of hand hygiene in disease transmission [4]. In 1884 Koch and Friedrich then formulated four criteria to establish a causative relationship between a microbe and a disease [5]. Thus, it was from this relatively recent evidence base that the germ theory of disease became the foundational concept of contemporary Western-based IP&C programs.

International IP&C guidelines are based on the relatively new (in human history) germ theory and even newer SENIC study. These international IP&C guidelines are assumed to be universally implementable across all healthcare settings to prevent and control infectious disease transmission. Comprehensive IP&C programs informed by these guidelines are both human-resource and financially intensive. Surveillance and reporting systems, antibiotic stewardship, performance improvement strategies, highly technical sterilisation and disinfection practices, environmental controls and the supply and use of optimal hand hygiene agents all form essential components of comprehensive IP&C programs [3]. Such IP&C programs have often been 'lifted' from well-resourced healthcare settings with an assumption that they can be directly placed into health care facilities anywhere in the world, regardless of different resources, language, culture or beliefs about disease causation and transmission. Lifting and placing these IP&C programs into different settings also assumes that health systems and staff are informed by—and base their work on–western scientific evidence and that the germ theory underpins decision making. Although fundamentally important, there is little evidence that these assumptions are systematically scrutinised before IP&C systems are transferred or implemented across the globe. However there is evidence that health systems in lower-middle income countries (LMICs) (countries with less than $1,230 US gross national income per capita) face fundamental challenges when attempting to implement international IP&C programs [6]. High burdens of communicable and non-communicable disease, poorly functioning or maintained health infrastructure and governance, high costs of treatment and local geographical and climatic factors make the implementation of high cost and intensively resourced contemporary western-based IP&C programs almost impossible in many resource limited settings [6].

There is clear evidence that different populations around the world, including those in LMICs, have a variety of beliefs about how sickness is caused, and diseases are transmitted. These beliefs may or may not align with the germ theory [7]. However there is little evidence on how health system leaders are investigating these beliefs and incorporating them into local IP&C programs. The influence that social, cultural and spiritual practices and beliefs have on health care workers (HCWs) and their IP&C practice is also poorly described. This is despite healthcare facilities being staffed by HCWs who serve their own communities and balance their own biomedical training (including the germ theory) with their community's underlying belief systems about sickness, disease and health [6]. In an attempt to find solutions to infection control challenges in LMICs, Sparke, Diau, MacLaren and West [6] found western-based IP&C programs embedded in countries where cultural, spiritual and religious beliefs were often at odds with the biomedical (germ theory) premise of contemporary international IP&C interventions. The lack of IP&C program success in these countries was exemplified by Allegranzi, Memish, Donaldson and Pittet [8], who poignantly illustrated that the use of alcohol-based hand rub (ABHR) for hand hygiene was still recommended in countries where alcohol is prohibited due to religious convention.

To apply sustainable IP&C guidelines and practices in LMIC with culturally, spiritually and linguistically diverse populations with any modicum of success, a new model of IP&C needs to be considered. Not one that is founded in the Western-biomedical model alone, but one which

also takes into consideration the cultural and spiritual context, knowledge and beliefs of disease causation and transmission, and one within which the biomedical model can sit.

## Culturally centred approaches to understanding infection prevention and control

People across the globe live with complex cultural beliefs and often do not live with a single reality, rather engaging with co-existing beliefs that transcend multiple belief systems. These belief systems may or may not include biomedicine. The current debates in countries across the globe on the cause and transmission of COVID-19, IP&C protocols that may or may not include face masks, and scepticism over COVID vaccination demonstrate that the germ theory does not uniformly inform peoples' beliefs about disease and prevention in every setting. A contagion, the central player in the biomedical model of disease transmission, is not recognised in the belief systems of many societies [7]. In many cultures sickness transmission could be seen to follow biomedical principles as the belief that if one person comes into contact with another person who is sick, that person will also develop the sickness. However the underlying belief of how this occurs is not necessarily attributed to the germ theory, but can be a complex mix of a culturally, ethereal and spiritually influenced knowledge base [7].

Beliefs around the cause of sickness also dictates treatment seeking practices. If it is believed that illness is caused by witchcraft or sorcery, then seeking help from health providers such as a traditional healer or Christian faith healer may be more likely [9], however if a germ is perceived as the causation of disease then a health service such as a hospital or health clinic may be the primary point of contact.

Most, if not all cultures adopt some form of medical pluralism where perceptions of illness are diverse and changing health providers between western biomedicine and traditional/folk medicine for healing is common [10]. Medical pluralism has been demonstrated to delay treatment, particularly for those seeking treatment in rural areas. Whilst expensive, a traditional practitioner is often the most frequent point of contact for rural communities due to distrust of western biomedical models and poor access to basic biomedical care [10]. Medical pluralism when treatment seeking is not always an ideological approach, but a pragmatic one. Often those seeking treatment see nothing wrong in using western biomedical and traditional healing strategies concurrently, and healers themselves often span multiple worlds, placing themselves strategically between different genres of therapy [11].

With vast differences in beliefs in the concept of sickness causation, transmission and treatment-seeking practices and the reality of medical pluralism, introducing an IP&C system based on the biomedical model alone will, by design, always ever be a partial solution to reducing infectious disease transmission. The complex mix of cultural, spiritual and Western knowledge within a population makes implementing IP&C systems based on the germ theory challenging. The implementation of a meaningful and sustainable IP&C program cannot simply follow an external 'blueprint' but requires an understanding of the local cultural context and an interpretation of the fundamental meanings that inform people's actions.

## Research context

The context for this study is Atoifi Adventist Hospital (AAH), located in a remote area of the Solomon Islands. AAH serves a population of around 80,000 people, many living in small remote villages, and is located in the East Kwaio region of the island of Malaita [12]. Around 95% of Solomon Islanders identify themselves as Christians, particularly those living in coastal villages, however many ancestral religious beliefs and traditional practices are still strongly held and actively practiced by those that live in the surrounding mountain hamlets [13]. For

the residents of the mountain villages, interacting with ancestral spirits are integral in everyday life as they are ever present, continually guiding their decision-making [14]. The success or failure in many everyday acts of living, such as crop production will only succeed with the support of ancestral spirits. The rules dictated by their ancestors and followed by the mountain people of Kwaio are complex, and a violation of these rules or 'tabu' brings punishment in the form of social and economic misfortune, sickness and death [13]. Likewise, the residents of coastal villages interact with a range of introduced Christian deities and spirits that include Jesus and his 'guardian' angels and Satan and his 'evil' angels. These deities and spirits are foundational to everyday life and guide decision making. The rules dictated by Christian leaders are complex and violation of 'Christian teaching' brings punishment from God and social and economic exclusion from villages.

The 65-bed AAH, opened in 1966, is operated by the Seventh Day Adventist (SDA) church in the Solomon Islands. AAH was designed by white SDA missionaries as an evangelical tool to convert people from 'heathenism' to become members of the SDA church. From the opening of the hospital in 1966, staff identified themselves as 'medical missionaries'. This terminology and approach continue today. Despite the early Missionary's exposure to the Melanesian culture of Papua New Guinea, the hospital design lacked input from the local community and as such presented many challenges for the Kwaio people's customary beliefs [15]. The layout of the hospital where the maternity ward and women's toilets is incorporated within the main hospital building does not reflect the Kwaio cultural systems or ways of maintaining a healthy individual, family or community. In Kwaio *Kastom* law it is *tabu* for a man to enter an area containing women's bodily fluids, thus breaking ancestral law, and as such many Kwaio who worship their ancestors do not access hospital services [12,16]. The services provided at AAH strictly follow SDA interpretations of Christianity including SDA food taboos that enforce Judaic rules of 'clean and 'unclean' foods. Despite these challenges for people who practice ancestral religion, the hospital has had a solid reputation for delivering high quality care to a predominantly Christian population.

In determining how IP&C practices could be improved at AAH to aid the development of an acceptable and sustainable IP&C program, the meaning of IP&C needed to be explored with staff and the community members which the hospital serves. Therefore, with the primary question being 'How can infection prevention and control practices be improved at AAH, Solomon Islands?', the following two questions needed to be resolved; What are current IP&C practice sat AAH? and can culturally informed interventions be developed and implemented to enhance infection control practices?

## Methods

Participatory Action Research (PAR) was used in this study because it provides a framework for community-based participatory research (CBPR) in partnership with a variety of people within this complex setting [17]. The participatory methodology was supported by a critical qualitative approach. Critical approaches can challenge assumptions and expectations of normal practice that have built over time. Often, HCWs have the impression that their work conditions, and the social and political injustices they face cannot be changed [18]. However critical theories aim to integrate theory and practice so that people become aware of discrepancies in their social practices and are motivated to change them.

The inherent premise of CBPR is that research is done *with* a community, not *on* a community and one where the community is encouraged to examine and analyse their circumstances. As such community members become co-researchers and work together to co-create knowledge, through research design and data collection and analysis [19]. The design of this study

was discussed with the Director of Nursing (DON) and the Chief Medical Officer (CMO) at the outset. Because the primary researcher was invited in to 'assist', not 'fix' AAHs IP&C issues, this brought with it the notion of participation. Further to this, they live with their reality, therefore input and expertise from the hospital staff was essential in the study design, and this also moderated the primary researcher's reflexivity. Solomon Islands hierarchical and social structures are complex to an outsider, therefore communication with the DON and CMO was vital to ensure data collection methods didn't break cultural and social boundaries. Co-researchers, who were mostly senior HCW, also had the opportunity to be participants in the study. Along with design input, co-researchers co-designed the Photovoice statement and organised the 'practicalities' of the study including participant recruitment and assisting with instructions around photography using a smart phone.

Photovoice was the primary method used to directly investigate how people understood disease transmission between people and how to prevent this transmission during phase one of the study, followed by Photo-elicitation in phase 2. Photovoice is a method of data collection, analysis and interpretation using photography [20]. Photovoice was developed in the 1990s by Wang and Burris [21] and is a qualitative action oriented research method [22] commonly used in education, health and social sciences. Photovoice is a method that provides cameras to research participants to take photographs based on a co-designed statement. Photographs are then reviewed in partnership between the research participants and the researcher to explore the story behind the photograph and how that story helps answer the research question. In Photo-elicitation, photographs are used to anchor dialogue about experiences of the participants [23]. Photographs can be taken by the researcher or the participants and in this study, photographs taken by 'other' participant groups (that is photographs used to generate discussion were not taken by the small group interviewees) to stimulate alternative views of the same situation. Photovoice and Photo-elicitation are foundational to CBPR because it values a re-balancing of power and trust between research participant and researcher and increases ownership of the research process and research results by research participants [24,25]. This method allows an in-depth investigation of the meaning behind the photographs to gain a richer understanding of salient issues because participants reflect on their own perspectives. Importantly photographs taken within a Photovoice study convey socio-cultural perspectives of the issue being studied [25].

The successful implementation of an IP&C program relies on the commitment of both trained HCW and 'ancillary' staff such as cleaners, laundry workers, kitchen staff and maintenance staff, all of whom play a pivotal role in hospital hygiene. Ancillary staff often have a lower level of formal education, meaning they may be of lower socio-economic status and have not been taught the same medical or technical terminology as healthcare workers. Ancillary staff are often employed from communities surrounding health care facilities and are often the same social and/or cultural group as patients of the facility [26]. Photovoice therefore provides an avenue for those who don't use medical or technical terminology and/or may have not been taught reading or writing to participate in the research process and have their stories and their 'voice' prioritised in a research project [21,27]. Providing a camera to HCW and to ancillary staff at AAH to document their concerns and understanding of IP&C removed the barriers of language, position, educational level and socio-economic status and enabled open discussion around IP&C problems and solutions.

## Data collection

Data collection occurred at AAH over two time periods. Phase one in October and November of 2018 and phase two 12 months later in October 2019. This allowed time between the two

phases for interview translation, transcription and data analysis. Participant validation occurred prior to the second round of interviews where participant photos were presented back to the participants, along with interview transcripts to verify the correctness of phase 1 interviews. Meeting with senior AAH staff, senior staff from the co-located Pacific Adventist University (PAU), and a research assistant (RA) (a co-author) who spoke fluent English, Solomon Islands Pijin and Kwaio (who translated information between participants and primary researcher throughout the data collection phases) created a list of HCWs and ancillary staff who played a pivotal role in IP&C at AAH. Initial recruitment consisted of 24 staff, 13 with professional biomedical education and 11 with lower levels of education or none at all; and two community members, one who received primary school and one with no education. Ages of participants ranged from 23 to 49 years including 4 participants with age unknown as they did not know their year of birth. The two community members were included at the recommendation of the Chief Medical Officer, to include perspectives from patients and their families about disease transmission. This provided a sample of different employment roles including ward nurses, nurse educators, medical laboratory technicians, pharmacy and Xray staff. Anciallry staff included cooks, cleaners and maintenance staff. The diverse sample of educational levels and village backgrounds offered alternate viewpoints about IP&C at AAH [28].

### Data collection procedure

Data collection followed the three common Photovoice components: Firstly training, that included the use of cameras, the ethics of photography, the misuse of photographic 'power' and the return of photographs back to the community; Secondly photo taking of objects or situations that participants believed best depicted the issue; Thirdly was participatory analysis. The participatory analysis had a three-part process which involved (i) selecting the most meaningful photographs, (ii) contextualising photographs by telling stories about the pictures, and (iii) codifying the photographs [29].

Following the training exercise, participant groups were given a camera (a smart phone) and a co-designed Photovoice statement. The statement designed by the primary researcher and co-researchers, was provided to the participants in both English and Solomon Islands Pijin and depicted the meaning of infection control and disease transmission in non-technical terminology. The English version 'From your perspective how can sickness pass from one person to another at Atoifi Hospital?' translated in Pijin to 'Lo tinting blo iu/iufela hao nao siki save pas go lo nara man taem stap lo hospitol?' The statement helped participants frame their photographs, provided a prompt, and also helped participants further understand the study [30]. Participant groups were homogenous work/professional groups as recommended by the co-researchers. Each participant group was given one camera for 48 hours and collectively asked to take no more than 30 photographs around the hospital that best depicted their perception of the Photovoice statement. When returned the photographs were downloaded to the laptop, the memory was cleared, and the camera was given to the next group.

Each homogenous work/professional group was interviewed about their photographs at a mutually convenient time and in a neutral or convenient place. Participants were provided with a printed and laminated copy of their photographs and jointly asked to choose 15 of their most meaningful, in cases where groups took less than 15, all photographs were discussed. The acronym PHOTO was used to guide the semi-structured group interviews:

P: Describe your **P**hoto

H: What is **H**appening in your picture?

O: Why did you take a photo **O**f this?

T: What does this picture **T**ell us about your role at the hospital?

O: How can this picture provide **O**pportunities for the hospital to improve?

[Adapted from 31]

The guide enabled participants to discuss the nuanced meaning behind the photographs, and allowed researchers and participants to explore components of the research question. Participants were asked to provide a caption for each photo which depicted the essence of the story. Explanation of the term 'caption' was required as the term does not have a literal equivalent in Pijin, and participants were given time to discuss the most meaningful caption for the chosen photos. The discussions were undertaken in a combination of Kwaio language, Solomon Island Pijin and English–and varied between groups. Field notes of interviewee responses were taken using the above framework to assist with data triangulation, and with a translator present, and interviews about the photographs were recorded using a digital audio recorder.

In phase 2 of the project (12 months after Phase 1), Photo-elicitation was used. It became apparent during the first phase of data collection that each participant held a different view of the same photo, therefore Photo-elicitation was used in phase 2 as a conduit for discussion. Photo-elicitation is where a photo is inserted into a research interview to evoke a deeper level of thinking about the object of the photo [32,33]. Each group were presented with all participant photographs taken in phase 1 (excluding their own), and they were reminded of the original Photovoice statement. Participants were given 10 minutes to choose five photographs that they thought best represented the statement. To elicit responses that best answered the research question, two questions were asked; from this photo, how does sickness pass from one person to another in the hospital, and what are the solutions? The differing educational, religious, work role and cultural backgrounds of the participants meant researchers did not assume that one person's perception of what causes sickness transmission in the hospital was a shared view.

Between phase 1 and phase 2 a major staff turnover occurred. Some staff moved to other hospitals in Solomon Islands and/or were involved in a major nurse recruitment drive to the neighbouring country of Vanuatu. This included the AAH Infection Control Nurse. As a result of staff turnover some new participants were recruited for phase 2. Their involvement was carefully considered as they were not involved in the initial photography in phase 1, however the technique of 'photo elicitation' did not require participants to comment on their own photographs [33]. New participants were valued because they offered their own valuable insights and their own interpretations of IP&C. As a result five newly recruited participants became a part of the study including two nurses and three maintenance staff.

## Data analysis

All photographs chosen by participants as the most depictive of the Photovoice statement, along with the associated interview transcripts and field notes were entered into NVivo software [34] and inductively analysed to explore initial themes/issues. This identified similarities and differences of conversations, perspectives, beliefs, concepts and actions. Nodes (codes), based on these emerging clusters were created using the NVivo software with sections of dialogue and related photographs allocated to nodes. Twenty-six nodes were initially identified in the first round of open coding. During the second cycle, codes were disassembled and reassembled according to their relationships with each other and collated under five broad themes (Table 1).

**Table 1. Themes emerging from data analysis.**

|  | Themes |
| --- | --- |
| **1** | What is known and believed about sickness transmission |
| **2a** | Knowing and practicing |
| **2b** | Knowing and NOT practicing |
| **2c** | Practicing and NOT knowing |
| **2d** | NOT practicing and NOT knowing |

Data from phase 2 were analysed and a further 15 nodes created which all 'fitted' within one of the five existing themes created during phase one. Using Photo-elicitation during the second phase enhanced the credibility and trustworthiness of the data as the analysis of the same photograph by different participants assisted with data triangulation [35]. Commonly chosen photographs indicated that the objects of the photographs were an issue not just for the initial photographer but more broadly across participant groups. In this cycle of data analysis 22 photographs were coded into nodes titled by the name of the photograph (eg. NEP11). Multiple photographs of the same object, taken by different photographers were coded to the same node.

To collate all the stories centred on a single photograph, from multiple participants, data from both phases which referenced the photograph was then cross-coded into the newly created node. Each photo node was then printed with their coding stripes, enabling a visual representation of the linkages between codes that occurred around a single photo across the entire data set.

Data aggregation resulted in two overarching themes. Firstly, what is known and believed about sickness transmission, and secondly, what the current IP&C practices at AAH are, and why. The second theme contained four subthemes (Table 1).

This paper presents and discusses results from Theme 1. Future publications will present and discuss results from Theme 2.

## Ethical considerations

This study received ethical approval from James Cook University (HREC—H7655), the Solomon Islands Health Research and Ethics Review Board (HRE No. 030/18) and a written letter of support from the Chief Executive Officer, Atoifi Adventist Hospital.

**Consent to participate.** Interviews were conducted after obtaining formal written consent from all participants involved in this study. Using Photovoice as the research method added an extra element to consent as it had to be clear that data collected, including photographs taken by participants may be used in research publications. This was written into the participant information sheet and consent form, and both were explained to participants in Pijin and the local language of Kwaio.

## Results

The multidimensional stories behind each photo made the analysis multifaceted. When visualising the relationships between the photographs, stories and themes, a tangled web was revealed. This web of ideas, concepts and themes covered some of the formal/routine/orthodox biomedical processes of IP&C, however, the Photovoice method enabled a much richer, nuanced and locally informed way of understanding IP&C at AHH.

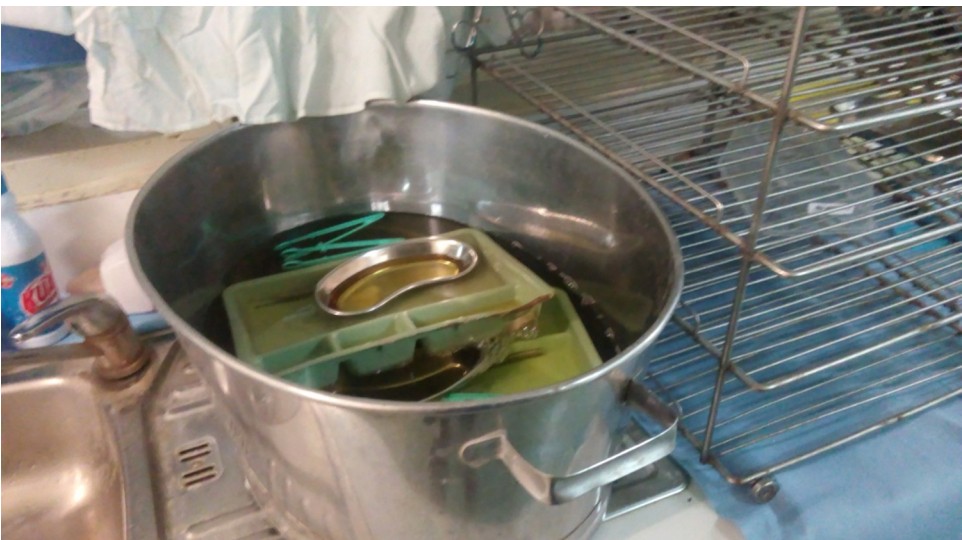

**Fig 1. JNP25 (Not proper soak inside).**

## What is known and believed about sickness transmission

Knowledge about sickness transmission were conveyed by photographs in two ways: (i) photographs explained through the germ theory and hospital hygiene processes; (ii) photographs explained through a social, spiritual and cultural process.

**(i) Photographs explained through the germ theory and hospital hygiene processes.** Sickness transmission was explained by a Junior Nurse through a photograph that represented inadequate disinfection processes of operating theatre surgical equipment (Fig 1);

> *The story here is that this is in theatre this is our soaking tray, so what happen is that. . . they bring it back, and we soak it here in this dish. Then after we soak it we wash it maybe after five minutes or fifteen minutes.*
>
> *After use we pour the dirty one. And we use the new one, but when we busy, sometimes. . .all the nurses just put it in the same water the dirty water and that's increase more germ in the water in the dish. And so if we don't come and see this one, maybe the whole day we just use the same water (JN1).*

A Senior Nurse explained sickness transmission through a photo of a dirty hand towel that represented how sickness could be transmitted by passing germs from one person's hands to another (Fig 2);

> *. . .The nurses we just use the same hand towel throughout the shift, and sometimes those coming in the other shift they use the same hand towel as well. To dry their hands after touching patient.*
>
> *I took this photo because it can cause infection to spread from one person to another, also the hand towel there have germs on the hand towels (SN1).*

Sickness transmission was explained by a cleaner who followed routine hospital hygiene processes. Although not specifically talking about germs or infectious agents the cleaner did

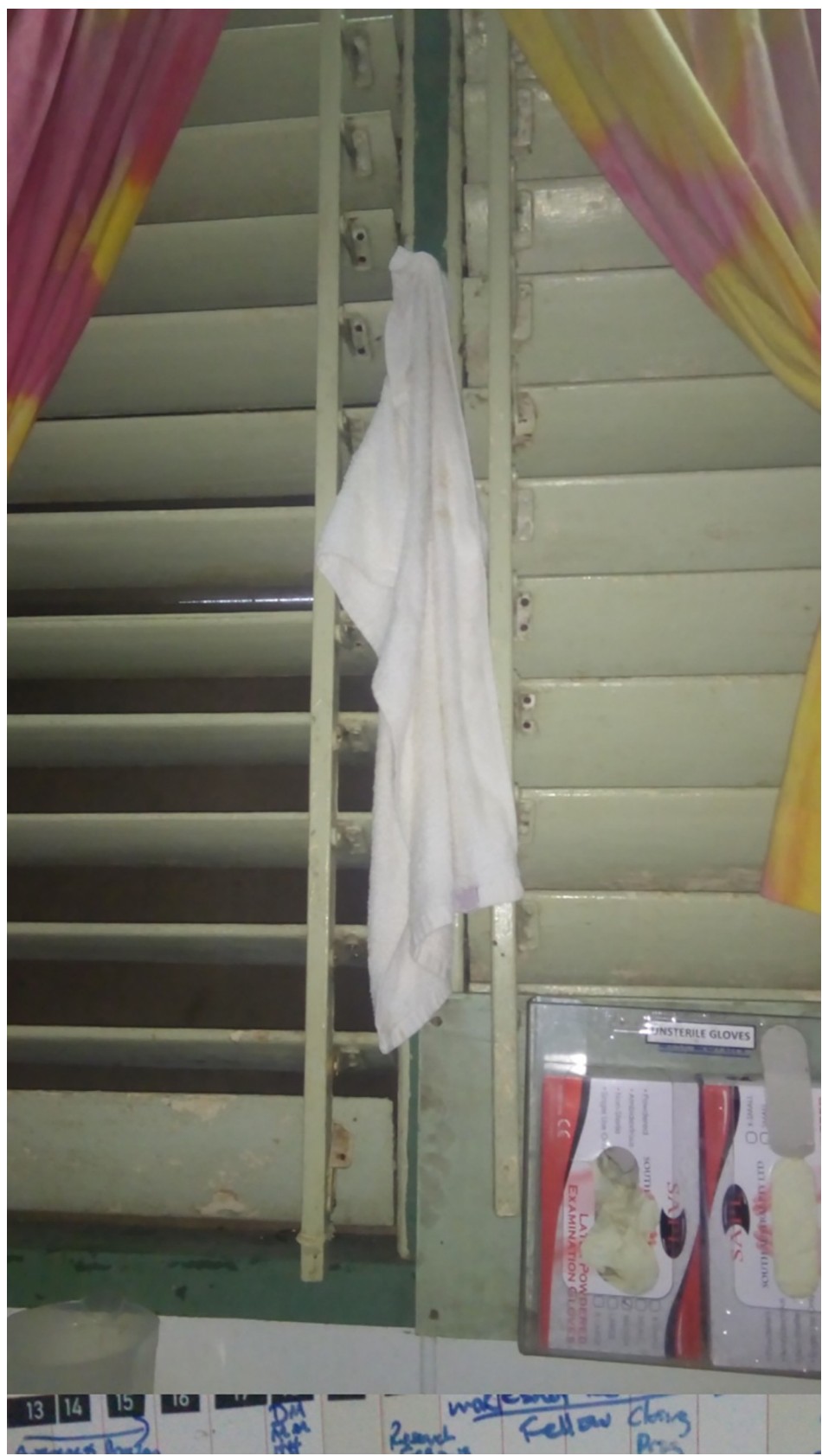

**Fig 2. SNP02 (Same hand towel, more hands, more infection).**

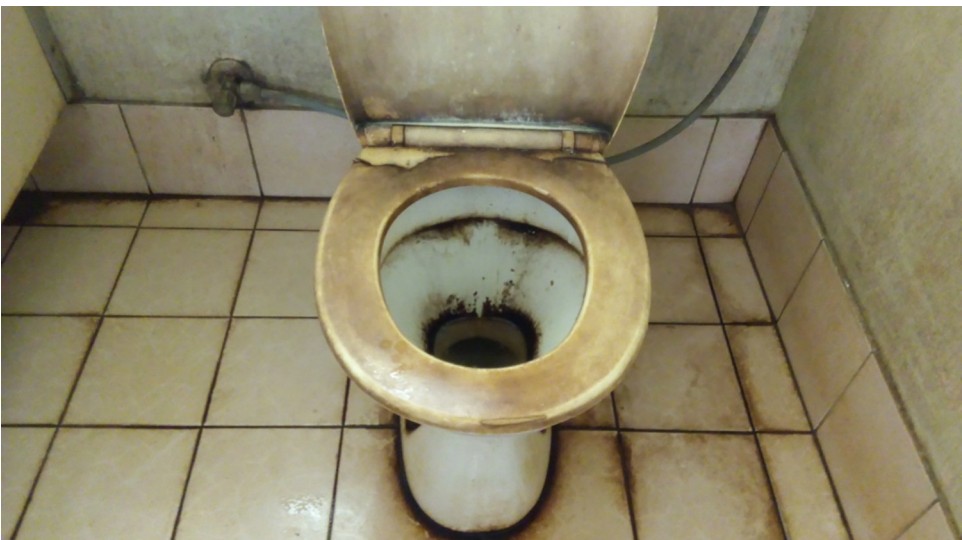

**Fig 3. CP5 (A dirty toilet).**

identify a photograph of a dirty patient toilet that represented how sickness could be transmitted when the toilet was not clean (Fig 3);

> *This is a toilet you see here, its have dirty on it, when the sick patient use this toilet, and then sometimes some of the patient doesn't use it properly, and it can pass sickness to another patient (C1—translated).*

A cook explained sickness transmission through a photograph that represented the importance of wearing a uniform to maintain hygiene practices that would prevent sickness transmission (Fig 4);

> *She say that, we have to wear uniform and clean, to cover our body, because body is dirty, so we have to cover our body, otherwise the dirty go in to the food, and when we use it and will getting sick (K1—translated).*

Sickness transmission was also explained by the maintenance staff who described sickness transmission when hospital hygiene processes were not followed through a photo of a rubbish bin without a lid (Fig 5);

> *The rubbish here smell, because the rubbish bin doesn't have any lid on it, and when its rotten the flies step in and go back like step on the food, and it will affect the patient and also it will cause diarrhea (W5—translated).*

**(ii) Photographs explained through a social, spiritual and cultural process.** Causes of disease transmission explained through social, spiritual and cultural processes were expressed by all participant groups regardless of their role, education or position at the hospital. Many participants, both HCW and ancillary staff, took photographs of rubbish around the hospital grounds and emphasised the influence of ancestral spirits on sickness transmission for themselves and patients. All participants were aware that people from the mountain villages believed the hospital rubbish contained female bodily fluids. When the rubbish was burned the smoke

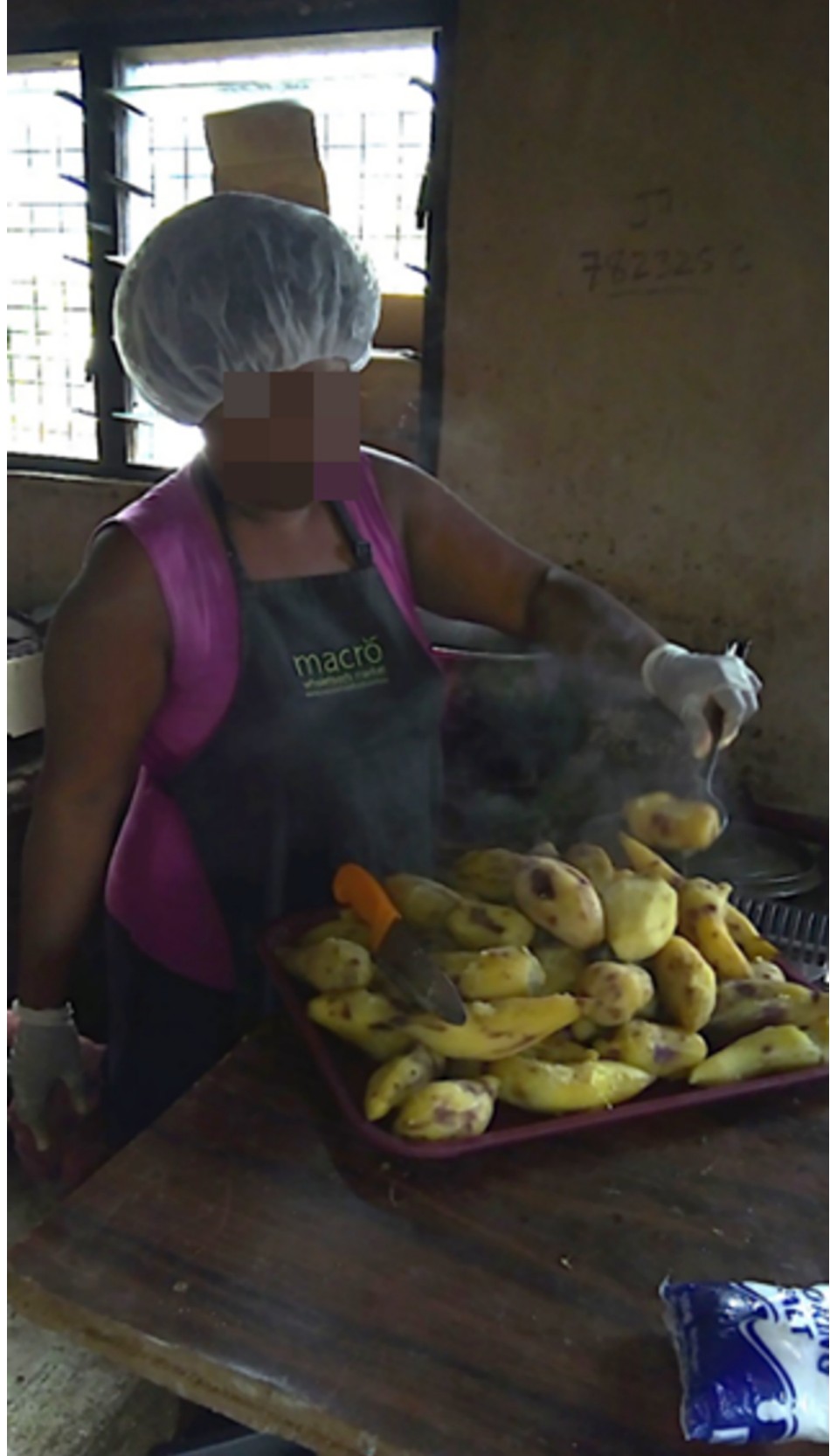

**Fig 4. KP18 (It's important to have uniform).**

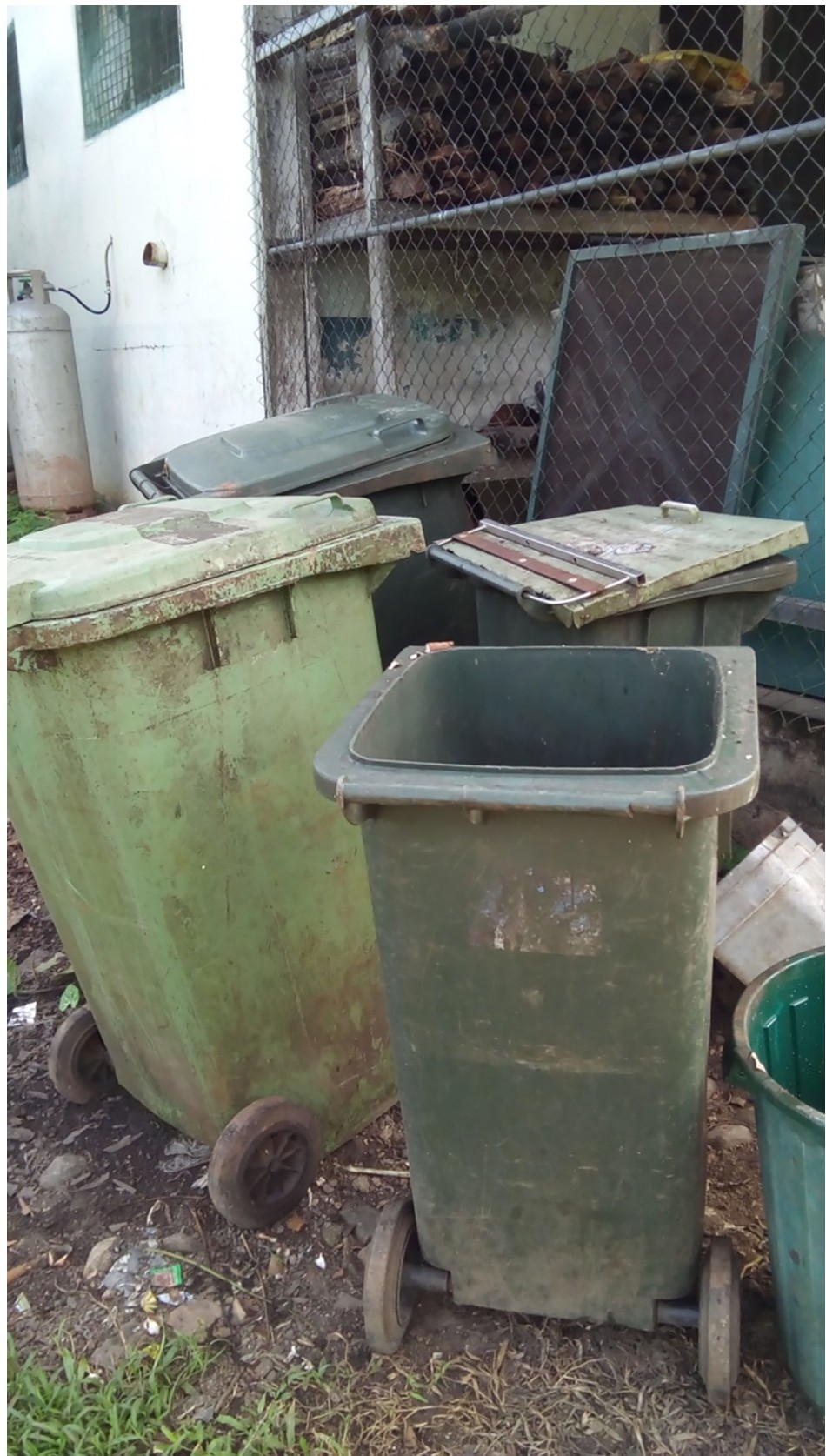

**Fig 5. WSP3 (No good, it's bad).**

from the burning female body fluids would enter the realm of the ancestral spirits and anger the spirits. People need to maintain relationships with ancestral spirits to ensure protection from harm and maintain good living. Because ancestral spirits were angered, protection could be withdrawn and people would become sicker or not recover from existing sickness (Fig 6).

*The pagan/custom or bush people. . .they don't want Maternity (sic) ward. . .when they burn the rubbish there, it will cause for the sick mountain people, will gettem sick (F2).*

Hospital staff were fully aware that regardless of the fact that a person is taking hospital medication, when the people from the mountains return home they are required to restore their relationship with ancestors by providing a sacrificial offering to appease their ancestors which will improve their health.

*Like when the bush people go back to their homes, they can sacrifice pigs for their Ancestors to make them better (F2).*

The maintenance staff responded to this explanation of disease transmission and implications for rubbish disposal sites;

*. . .the problem is some mountain people they came down and say something they go "oh they burn something from the rubbish", and they will ask compensation on the hospital so, even we put the incinerator down there, they still talk about it, so we have no option to throw the rubbish so we go down strait to the wharf (W1).*

Education about effective disease transmission prevention is hindered due to the opposing belief systems between the biomedically trained HCW and the spiritual believers from the mountains. Despite knowing about the spiritual beliefs that have underpinned the mountain people's society and culture for thousands of years, a Junior Nurse intimated that the mountain people will change their beliefs to the biomedical model if it is explained to them;

*So if someone came from the hospital and they're from the mountains, do you think they understand the cause and effect of diseases (Researcher)?*

Some do, some don't. . . But I think if we explain clear to them, they will understand (JN1). Some participants explained sickness transmission akin to Miasma theory;

*The flies sometimes step on our food, the flies came and the rubbish getting rotten and smell and the worm get into the rubbish. It's easily to affect us patient. . .when the patient smell it and breathe it, and get into her body (F1—translated).*

Causes of disease transmission were not only explained through traditional spiritual worldviews, but also through the introduced religious system of Christianity. The Medical Laboratory Technologist (MLT) with university degree in biomedical science explained disease transmission and infection prevention through his belief in the protection from the Christian God when the biohazard cabinet designed to protect him from infectious agents in the hospital laboratory broke down.

*Well, Am feel discourage, but you know working in institutions like this, I would say owned by the church, so our mind say, thinking the lord will protect me from this. From this*

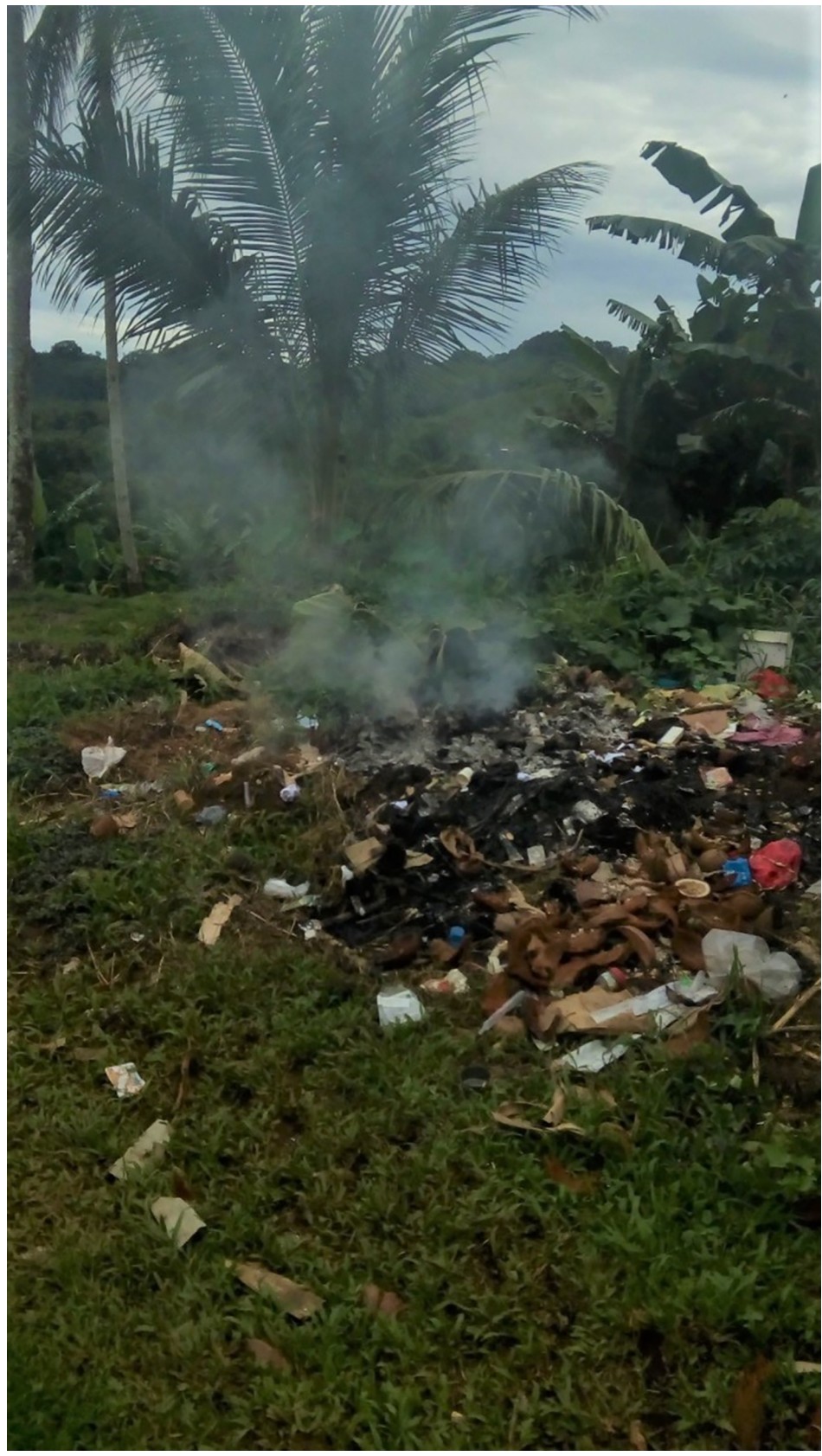

**Fig 6. F2P3 (Polluted air).**

*things. . ..So I think He will definitely protect me, because I am doing people good thing to serve people. That's the concept behind my thinking (Pa2).*

Causes of disease transmission were also explained through social connections and kinship relationships. Social relationships determined by local Melanesian culture was commonly described as an underlying influence contributing to sickness transmission and facilitating and/or preventing disease transmission;

*. . . we think of this infection prevention and control is very important it's seriously important for us to consider, that perspective then drives each of us to be seriously like helping each other. But as Melanesian cultures sometimes we feel that like if somebody talks to us directly it gives us bad feelings like that. Sometimes our culture contributes to that (NE1).*

The photograph of a potato on a plate in a share-house that had been gnawed by a rat (Fig 7) provided insight into Solomon Island kinship obligations which impact HCW practice. When members of the share house became ill as a result of eating the potato, the pharmacy assistant, a 'kin' through blood relations to other household members, was obliged to supply medications to her housemates; medications intended for patients.

*Like this one, the staff we live here so when we sick, there will be no workers for the hospital, and also when we sick the drug will be use. . .I feel very angry sometimes. Because I control the drug and I was just thinking, people should stay hygienic, careless, they come and take the drug that should be given to the people here in the hospital (Ph1).*

Cultural traditions and expectations were also used to explain why medical supplies hadn't arrived on the monthly barge (there are no roads to the hospital on the remote east coast of Malaita and so all supplies come on the monthly barge). This was explained to impede supply and influence sickness transmission;

*I place my order but they don't send it in the ship, so we're running low until now. . . there's one lady, her husband is very sick and he's died, so every workers they go and attend the funeral so they can't do my order (Ph1).*

Finally, the pharmacy assistant described the differences of personal hygiene and the differences between hospital and village toilets as a cause of sickness transmission;

*Like we don't know, mainly here in Solomons, native people they go to the toilet, they don't have proper [flush] toilet, and some just use leaves to wipe with their backdoors, and some go to the sea, and sometimes the patient they have like long nails, and some. . .like use their hands to wash their backdoors when after go to the toilet (Ph1).*

## Discussion

Knowledge and beliefs around sickness causation and transmission in this setting has multiple layers. An attempt to improve IP&C based on international imported protocols informed exclusively by the germ theory may appear straight forward but this study provides evidence of deeper contextual factors that need to be considered. Consistent with the evidence from similar settings, this study found that relying on the germ theory alone as the underpinning epistemology for disease transmission may not have the desired outcome to prevent and control

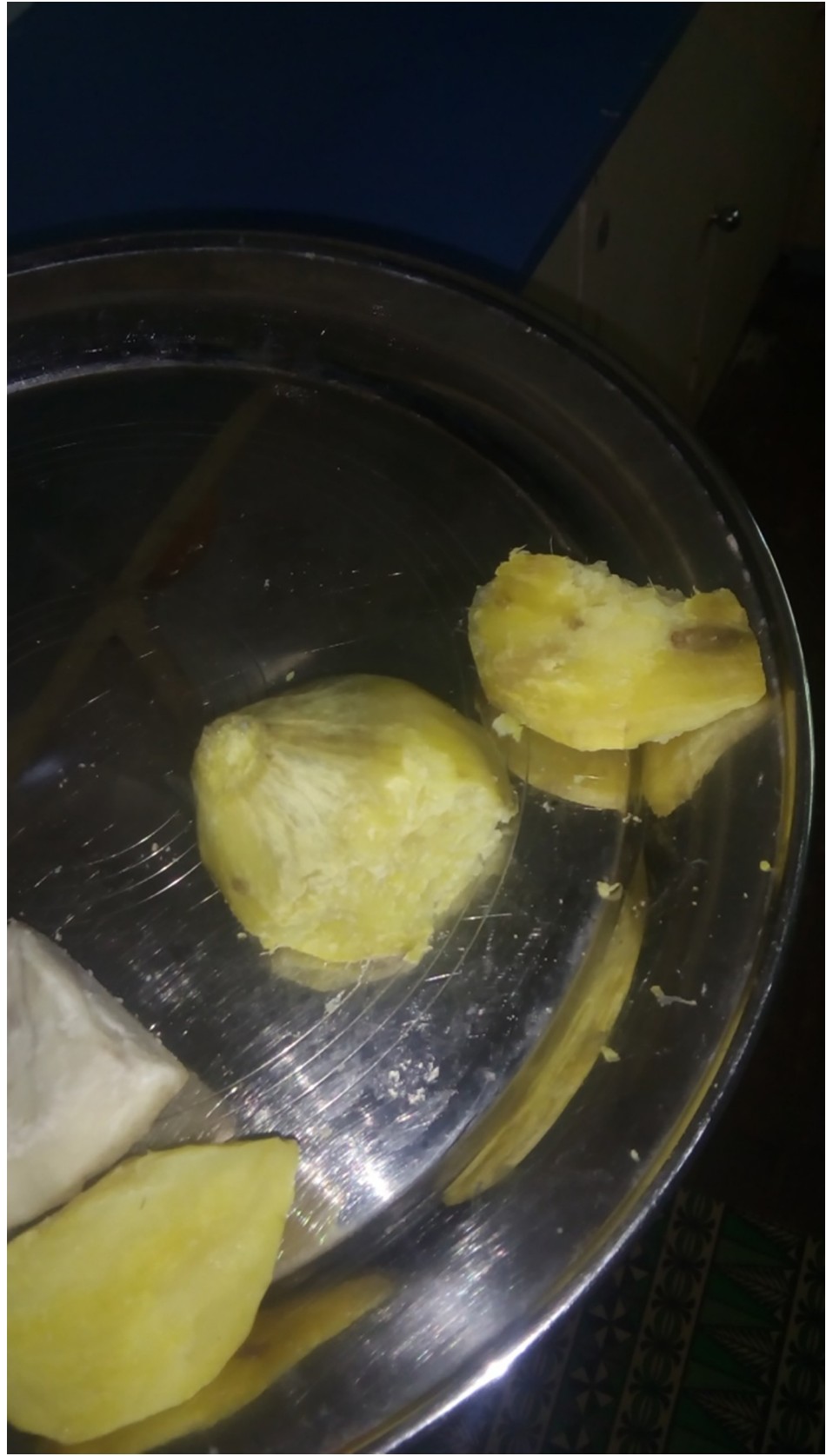

**Fig 7. PhP4 (The potato and the rats).**

 

infections. Traditional and contemporary beliefs about sickness causation always needs to be considered from a broader cultural perspective [36,37]. Some studies have described the influence that religious and cultural convention have on aspects of IP&C such as hand hygiene [8, 38] and blood borne pathogen transmission [26]. However, there are only very few studies investigating how spiritual and cultural beliefs impact HCWs perception of infection transmission more broadly. In some cultures, strong social hierarchy and status influences healthcare practice. Lower order tasks, many of which involve potential infection transmission, such as removing intravenous cannulas [39] and cleaning body fluid-soaked sheets are delegated to cleaners and family members as they are considered by nurses to be below their social status [40]. This study is one step towards filling the knowledge gap between cultural and spiritual influences on perceptions of infection transmission and the impact for HCW practice.

Most participants stated a belief that some form of physical entity rather than an ethereal one was involved in sickness transmission. Participants with biomedical training indicated that germs were a causative organism. When describing their photographs of a dirty hand towel the Senior nurse said the dirt on the towel could transmit germs and the Junior Nurses described the re-use of the dirty instrument soaking water in theatre causing an increase of germs. Non biomedically educated staff such as cleaners expressed human faeces or hospital rubbish as a mode of transmission yet didn't articulate a specific organism. Disparities in IP&C education is not uncommon in low- middle-income countries, with ancillary staff such as cleaners being left out of educational activities, yet expected to undertake hygiene procedures [39]. Interestingly in this study, despite lacking biomedical knowledge, cleaners knew that 'sickness' could be transmitted though fomites, and wore PPE accordingly. The belief that a physical entity could transmit disease was not uniform as some participants described sickness transmission from multiple perspectives. Rubbish was a concern for many participants as they explained sickness transmission through the feet of flies, but also through the smell of the rubbish. When improving IP&C processes at a hospital such as AAH, the germ theory should not be cast aside. People explicitly described practices such as cleaning the toilet, soaking surgical instruments and changing hand towels as important to prevent sickness transmission demonstrating the belief that 'something' transmits sickness.

This study also identified that, similar to other remote hospitals in LMICs, AAH had limited human, physical and financial resources to address IP&C [41]. Resource limitations in any setting will ultimately lead to IP&C breaches and potential infection transmission. The ability of this study to delve into 'the causes of the causes' through Photovoice, showed that cultural and societal expectations can be the cause of supply issues, even when there are no financial constraints. Cultural obligations and kinship relations made it difficult for the pharmacy assistant in this study to manage the hospital medications appropriately.

Kinship relations and cultural responsibilities including the general rules of respect, obligation, mutual exchange and social property, govern much of the cultural construction of Solomon Islanders [42]. Many family members work in various roles within the hospital, and kinship laws and cultural custom make it difficult for some staff to hold others to account. Expressing a professional and educated biomedical opinion such as reminding your kin about proper IP&C practice is problematic as it can break kinship rules and has the potential to lead to poor practice and infection transmission.

A compounding influence on the staffs' capacity to exercise their positional responsibilities is the theocratic governance of the hospital. As a SDA administered health facility, the hospital leader, the Chief Executive Officer (CEO), is required to be a member of the SDA Church and guided by the church's doctrinal position [43]. Staff of AAH are employed by the Church and are guided by direction from the CEO. Therefore a dichotomy exists between the biomedical basis of disease transmission and prevention and the Christian call for God's healing.

Managerial role modelling is an important motivational factor in providing safe infection control practices [44]. Thus, for staff with a biomedical background such as the medical laboratory technician, IP&C practice is a juxtaposition of axiological values of science-based professional education and the religious beliefs that need to be followed because of the requirement to follow the theocratic management and governance of the hospital.

Ancestral spiritual beliefs play a significant role in the mountain people's decisions about attending AAH, as the built environment of the hospital is not conducive to their beliefs [15]. The *wantoks* and the maintenance staff openly discussed the *tabus* associated with the rubbish. The mountain peoples' ancestral beliefs are known by Christian staff members however the conflicting belief systems make it hard for staff to understand and accept, which then influences the IP&C messages provided to patients. Using germ theory to educate the mountain people to wear footwear will, according to the Junior Nurses, make the mountain people compliant. Conversely, the maintenance men understood the belief systems and knew that digging a rubbish pit by the side of the road was not an acceptable option for the mountain people. They therefore found an alternative, albeit one that is deemed poor practice from a formal IP&C perspective.

The biomedical model of disease causation and transmission in IP&C should not be excluded to make way for spiritual, cultural and religious world views. However, the vast differences in sickness causation and transmission beliefs should provide a foundational view point that becomes incorporated into the planning and implementation of IP&C programs in settings such as AAH. Further research needs to be undertaken on understanding the interplay between the germ theory, societal and local culture, ancestral spirits and Christianity and how these influence sickness transmission knowledge and beliefs to provide the underpinning concepts for a workable, pragmatic and locally appropriate IP&C program to prevent infectious disease transmission in this and similar settings.

## Limitations

The results of this study only reflect the experience of one group of people in one hospital in one country, and therefore despite cultural similarities, are not reflective of the whole of the Solomon Islands or other Pacific Island nations. Given the IP&C expertise of the lead author, there was the potential to focus only on the biomedical basis for disease causation and transmission. However, co-researchers and co-authors ensured reflexivity was considered throughout the study to mitigate this potential bias.

## Conclusion

This study is revealing of participant's epistemology, that is what they know about disease transmission through societal norms, religion, formal education and cultural influences and beliefs. Expecting uniformity across culturally and spiritually diverse populations, and to directly replace their beliefs and behaviours regarding sickness causation and transmission with measures informed by the germ theory does not reflect reality. Researchers can not ignore the structural nuances of culturally and spiritually different health contexts and expect them to conform to Western hierarchies therefore IP&C researchers need to change how evidence is viewed. Paying credence to cultural connotations and implications and investigating a community's and individual's knowledge systems around disease causation and transmission is required to understand the context in which IP&C is being implemented. Without this IP&C will keep ending up in the same place, with no progression.

## Acknowledgments

The authors would like to acknowledge the staff and community of Atoifi Adventist Hospital for allowing this study to proceed, and to the co-researchers who informed the study design, data collection and analysis.

## Author Contributions

**Conceptualization:** Vanessa L. Sparke, Caryn West.

**Data curation:** Vanessa L. Sparke, Dorothy Esau.

**Formal analysis:** Vanessa L. Sparke.

**Investigation:** Vanessa L. Sparke, Dorothy Esau.

**Methodology:** Vanessa L. Sparke, David MacLaren, Caryn West.

**Project administration:** Vanessa L. Sparke, Dorothy Esau.

**Resources:** Vanessa L. Sparke.

**Supervision:** David MacLaren, Caryn West.

**Writing – original draft:** Vanessa L. Sparke.

**Writing – review & editing:** David MacLaren, Dorothy Esau, Caryn West.

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
