## [Decision Letter · Decision Letter 0]

17 Jun 2022

PGPH-D-21-00961

Changing the lens through which we see others and the world: Infection prevention and control insights using Photovoice

Dear Dr. Sparke,

Thank you for submitting your manuscript to PLOS Global Public Health. After careful consideration, we feel that it has merit but does not fully meet PLOS Global Public Health’s publication criteria as it currently stands. Therefore, we invite you to submit a revised version of the manuscript that addresses the points raised during the review process.

We look forward to receiving your revised manuscript.

Kind regards,

Mahbub-Ul Alam, MPH

Academic Editor

Journal Requirements:

1. Please include a complete copy of PLOS’ questionnaire on inclusiveness in global research in your revised manuscript. Our policy for research in this area aims to improve transparency in the reporting of research performed outside of researchers’ own country or community. The policy applies to researchers who have traveled to a different country to conduct research, research with Indigenous populations or their lands, and research on cultural artifacts. The questionnaire can also be requested at the journal’s discretion for any other submissions, even if these conditions are not met.  Please find more information on the policy and a link to download a blank copy of the questionnaire here: https://journals.plos.org/globalpublichealth/s/best-practices-in-research-reporting. Please upload a completed version of your questionnaire as Supporting Information when you resubmit your manuscript.

2. Please update your Competing Interests statement. If you have no competing interests to declare, please state: “The authors have declared that no competing interests exist.”

3. All figures and supporting information files will be published under the Creative Commons Attribution License (creativecommons.org/licenses/by/4.0/). Authors retain ownership of the copyright for their article and are responsible for third-party content used in the article. 

Figures 2 to 8: Please confirm (a) that you are the photographer; or (b) provide written permission from the photographer to publish the photo(s) under our CC-BY 4.0 license.

Please upload any written confirmation as an 'Other' file type. It must clarify that the copyright holder understands and agrees to the terms of the CC BY 4.0 license; general permission forms that do not specify permission to publish under the CC BY 4.0 will not be accepted. Note that uploading an email confirmation is acceptable.

Additional Editor Comments (if provided):

Reviewers' comments:

Reviewer's Responses to Questions

**Comments to the Author**

1. Does this manuscript meet PLOS Global Public Health’s publication criteria? Is the manuscript technically sound, and do the data support the conclusions? The manuscript must describe methodologically and ethically rigorous research with conclusions that are appropriately drawn based on the data presented.

Reviewer #1: Yes

Reviewer #2: Partly

Reviewer #3: Yes

2. Has the statistical analysis been performed appropriately and rigorously?

Reviewer #1: N/A

Reviewer #2: N/A

Reviewer #3: Yes

3. Have the authors made all data underlying the findings in their manuscript fully available (please refer to the Data Availability Statement at the start of the manuscript PDF file)?

Reviewer #1: Yes

Reviewer #2: Yes

Reviewer #3: No

4. Is the manuscript presented in an intelligible fashion and written in standard English?

Reviewer #1: Yes

Reviewer #2: Yes

Reviewer #3: Yes

5. Review Comments to the Author

Reviewer #1: (General) This article explores perceptions of infection prevention and control among staff and community members in a hospital in the Solomon Islands. The main conclusion is that germ theory, which underlies typical infection prevention and control protocols, should not be the only theory informing IPC guidelines and practical implementation. Instead, broader cultural and religious beliefs should also be considered in this context to integrate local knowledge and belief systems.

(General) The article presents the results of original research and was, for the most part, clear and easy to read. The participatory action research method undertaken is a novel and interesting one and was described in sufficient detail within the manuscript. The data collection procedure has also produced results that feed into the discussion and conclusions adequately.

The article makes an interesting contribution to the topic. In saying this, please find the following suggestions that may further improve the article:

1. (Major) Literature speaking to this exact issue seems to be lacking in the current, available literature, however, the paper could be further strengthened by literature speaking to similar issues around the influence of cultural and social beliefs on IPC. For instance, the article could integrate literature on barriers and facilitators to healthcare workers' adherence to IPC guidelines: https://www.cochranelibrary.com/cdsr/doi/10.1002/14651858.CD013582/pdf/full. Similarly, how cultural beliefs may influence HAI in other (institutional, geographic etc.) settings such as the "clean care is safer care" belief presented in this article could be reviewed: https://journals.plos.org/plosone/article?id=10.1371/journal.pone.0140509. There are other articles speaking around this topic and a more exhaustive review of the literature, particularly in the "Culturally centred approaches..." (line 103) section, would further strengthen the introduction/background and discussion sections of the article.

2. (Minor) The title is a bit too long and too general. I suggest that the first line before the colon be removed and replaced with something speaking to the use of Photovoice and the research being conducted in the Solomon Islands. This would indicate to the reader that the article explores the topic in one specific geographical context and using the particular method of Photovoice more explicitly.

3. (Minor) I would suggest that the authors re-read the manuscript to identify other typographical/grammatical errors but a few that I have identified are on:

• Line 106 – “engage” should be “engaging”

• Line 124 – “any all” should just be “all”

• Line 160 – “covert” should be “convert”

• Line 178 – could abbreviate infection prevention and control as done throughout the rest of the article

• Line 532 – should this line read “to NOT attend”?

• Line 538 – this sentence does not read very clearly as there is a full stop in the sentence beginning with “On the contrary…”

4. Line 425- (Major) The finding reads: “…mountain people will understand about medical model if it is explained to them”. In light of this, is the issue more about rethinking culture in health-related communication or about adapting IPC to be more culturally sensitive? I suggest that this quote is unpacked in greater detail and that the point is made more explicit.

5. Line 436- (Major) It would be useful to include what the authors see as the difference between “traditional spiritual frameworks” and “spiritual systems” by adding more information here.

6. Line 456 – (Major) It is not very clear how the potato and rat finding is an example of how “Melanesian culture reappeared as a reason for staff sickness”. If there is a reference to how the potato and rat are an example of Melanesian culture in the literature, I suggest that more information is added in this section. If not, perhaps this section should be reframed to make the finding more clear.

7. Line 488 – (Minor) It is indicated that other studies have been done in this area in other settings. The references for these studies should be included in this sentence.

8. Line 494 – (Major) It is mentioned that there are "only very few studies investigating how spiritual and cultural beliefs impact HCWs perception of infection transmission and subsequently how these perceptions influence practice." I suggest that these studies' findings are described in more detail here and that references be included.

9. Line 498-502 – (Major) It is not very clear what is meant by “physical entity” in this section. What “other spiritual entities” are being referred to here? What is the significance of highlighting this separateness/independence of the germs, faeces etc. in this context? Additionally, which “cultural processes” are being referred to? This section is a little vague; some conceptual clarification would provide greater clarity as to what exactly the point is that is being made in this section.

10. Line 527 – (Major) Beginning from “For staff…”. It would be useful to include why the "juxtaposition of axiological values" is an important consideration in this setting in the Discussion section. As it currently stands, it is mentioned very briefly but is a critical aspect of the central finding in this study: that it is important to consider cultural, social and religious beliefs for IPC. Further literature on this could be integrated to strengthen this important point. Grant and Cain's article on how healthcare workers construct spiritual meaning to bridge science and religion can be found here https://onlinelibrary.wiley.com/doi/full/10.1111/jssr.12285?casa_token=T3pJOyxcqdcAAAAA:xV4Iv8tLTtfffotGw9s4jPS0CC3_ZeQJsjI-ygSdwWr4d8W6TOc-CHPyqo5pG5iZDgfjVpxi18GfS14U. Other authors have also written about this important debate; tying the findings to this literature will greatly improve the Discussion section and highlight the less obvious findings that have been presented in the Findings section.

Reviewer #2: Review Comments to the Author

The manuscript describes a study that compellingly challenges assumptions about the implementation of international IP&C guidelines, especially in LMICs. This is done through a qualitative investigation of IP&C implementation in one hospital in a remote region of the Solomon Islands. The goal of this PAR is to inform models of IP&C that account for cultural, spiritual, and structural contexts.

Introduction & Rationale

The introduction comprehensively covers important content areas of relevance to the study and provides a good description of the context.

The conceptual orientation is less well described. Starting P5. Line 125. The binary positioning of ‘traditional/folk medicine’ with ‘modern’ (presumably western biomedicine?) is problematic and oversimplifies medical pluralism. Delete the word ‘both’ and replace ‘modern’ with western biomedicine if this is what is meant here. Suggest some nuancing of the discussion of medical pluralism, with perhaps some more specific examples from the literature, to show how different peoples and contexts negotiate multiple beliefs about health and infection prevention in a hegemonic biomedical (and possibly also evangelical Christian) context.

No explicit theoretical orientation is detailed for this study. Given the focus on culture and health, did concepts from critical medical anthropology, for example, inform the study? Other critical theory? If so elaborate.

The hospital history provided is compelling and very interesting, with staff identified as medical missionaries. There are then two overlapping hegemonic paradigms of health and disease transmission operating in this setting -biomedical (IP&C) and Evangelical Christian. It is not clear how the study grappled with the distinctions and intersections and how they might play out in association with client/patient population beliefs about health and experience accessing services.

This section should conclude with a clear statement of the research objective and any associated research questions.

Methods

There is limited elaboration of the CBPR context of this study. Who were the partners? What was their role in the study at all stages? Where did the idea for the study come from? What was the role of the lead author/ researcher? Other authors on the paper?

Photovoice methods are described here and well rationalized. Later, in the section on data collection, photo elicitation appears. The method should be described and rationalized in the methods sections as well.

Two phases of data collection approximately a year apart are discussed but not explained. What was the purpose of the two-phase approach? Why these two time periods? Why photovoice in one phase and photo elicitation in the second?

The grouping of participants by role is rationalized. The level of detail about each individual participant in each phase of the study presented in tables is not needed and has enough information that it would be possible to identify individual participants and link some of them with the quotes presented in the results. Instead, consider summarizing participant information more generally, perhaps according to the groupings used in analysis.

Data Collection

Elaboration on the implementation of photovoice is needed. Participant groups were given cameras for 48 hours and asked to take no more than 30 photographs. Was each person in the group given a camera and each asked to take 30 photos? Or was a camera distributed among group members for a period of 48 hours, with a collective set of 30 photos produced? Group members were then brought together. Did they collectively choose the 15 photographs for discussion? Were the captions collectively determined? What logistical or other challenges were navigated?

For the photo elicitation phase, elaborate on how this worked. Even though the participants appear to have been different, were the groupings the same? What is meant by the statement (P12. Lines 274-75) that groups were presented with all of the photographs (excluding their own) and asked to choose 5 for discussion? If this group didn’t take photos, what was considered ‘their own’? What was excluded and why? How many photographs were participants asked to sort through in the allotted time period?

The data collection section identifies that project data include photographs and interview transcripts. A reference to field note data appears in a different section of the paper (analysis section), but is not explained or rationalized.

Data analysis

Some elaboration is needed. Who participated in the analysis? What was the role of co-authors? Community partners? Which photos were included in the NVIVO analysis? Rationale? Inclusion/exclusion criteria? Given that data was collected by groups, was there within and between group analysis? Why/why not? How were content, context, and conceptual orientation included in analysis or interpretation? Was gender attended to? Why/why not?

Results

Figure 1 is difficult to read and does not add much more information than is already provided in the description that precedes it in the text.

Some orientation to the presentation of findings under theme 1 (the focus of this paper) would be helpful. Are the headings in this section sub-themes? Interpretive?

Good use of quotes.

Some questions about the results: What explanation is provided for inadequate disinfection, such as dirty water, reuse of hand towels, uncleaned toilets (P15-16, Lines 354-360; 366-377; 375-77), and other breaches of hygiene practices? How did participants answer the opportunities questions?

P19. Lines 456-457. It is not clear how social relations played out as a reason for staff sickness. Where the social relations appear to come into play is in the obligation to treat sickness.

It is not clear how the quote about toileting practices (P.20, lines 478-482) is explained as a cultural difference. What makes this ‘cultural’ rather than contextual?

Discussion

Ensure no new uncited findings are brought up in the discussion E.g. P22. Lines 527-530.

Given this is PAR, some discussion of the action implications at a practical level would strengthen the manuscript. What opportunities do participants identify for biomedical issues identified? For cultural/spiritual issues? Is there support for or resistance to an IP&C strategy that attempts to accommodate the health and disease beliefs revealed in this study? What role does the ‘medical missionary’ context play in possible opportunities?

It would be interesting to discuss some of the overlaps that may have been revealed, for example, rubbish handling may have implications for the biomedical and cultural/spiritual perspectives on disease transmission.

Publication ethics considerations.

Elaborate individual and collective consent, especially with reference to the photographs. In what ways did participants and the community-based partner agree to the use of the photographs? For data analysis purposes? For use in academic or public presentations or publications? Was there a photo release process? Example: The person in figure 3 is potentially identifiable. The pixelated face suggests that this person may not have consented to be identified. Clarify photo release circumstances.

Other methodological and interpretive considerations

Strengths, challenges, and limitations are not identified or discussed.

No qualitative rigour criteria are identified or discussed with reference to this study.

Edit suggestions.

P2. Line 46. Cite SENIC

P5. Lines 105-107. Cite examples of this statement.

The paper could benefit from an edit review.

Some of the errors noted:

P4-5. Lines 99-102. Beginning “Not one that is found…” Incomplete sentence

P5. Line 112. Insert comma after ‘transmission’

P5. Lines 113-115. Awkward sentence.

P5. Line 124. Delete ‘any’

P5-6. Lines 125-127. Beginning “Medical pluralism…” Incomplete sentence

P7. Line 177. Following ‘members’ delete ‘to’

P15. Line 336. Change ‘was’ to ‘were’.

P23. Lines 538-539. Beginning “On the contrary…” Incomplete sentence

P23. Lines 559-560. Beginning, “Nor can researchers...” Incomplete sentence

Reviewer #3: Thanks to the authors for choosing an interesting method regarding Infection Prevention. However, I think the abstract does not reflect the results well. hence, this should be rewritten.

In the method section, the study site and study design should be explained with more clarity. Table 3 is mentioned in the data analysis section but is presented in the result section. This should be corrected.

The result section is comparatively well written, however, each domain or theme needs to be supported by facts, which was not found in all the cases.

The findings in the discussion, in some occasions, were not compared with the literature available. The authors are advised to have a recheck on this.

6. PLOS authors have the option to publish the peer review history of their article (what does this mean?). If published, this will include your full peer review and any attached files.

**Do you want your identity to be public for this peer review?** For information about this choice, including consent withdrawal, please see our Privacy Policy.

Reviewer #1: No

Reviewer #2: No

Reviewer #3: No

---

## [Decision Letter · Decision Letter 1]

7 Nov 2022

Exploring Infection prevention and control knowledge and beliefs in the Solomon Islands using Photovoice

PGPH-D-21-00961R1

Dear Ms Sparke,

We are pleased to inform you that your manuscript 'Exploring Infection prevention and control knowledge and beliefs in the Solomon Islands using Photovoice' has been provisionally accepted for publication in PLOS Global Public Health.

Best regards,

Julia Robinson

Executive Editor

Reviewer Comments (if any, and for reference):

Reviewer's Responses to Questions

**Comments to the Author**

1. If the authors have adequately addressed your comments raised in a previous round of review and you feel that this manuscript is now acceptable for publication, you may indicate that here to bypass the “Comments to the Author” section, enter your conflict of interest statement in the “Confidential to Editor” section, and submit your "Accept" recommendation.

Reviewer #1: All comments have been addressed

Reviewer #3: All comments have been addressed

2. Does this manuscript meet PLOS Global Public Health’s publication criteria? Is the manuscript technically sound, and do the data support the conclusions? The manuscript must describe methodologically and ethically rigorous research with conclusions that are appropriately drawn based on the data presented.

Reviewer #1: Yes

Reviewer #3: Yes

3. Has the statistical analysis been performed appropriately and rigorously?

Reviewer #1: N/A

Reviewer #3: Yes

4. Have the authors made all data underlying the findings in their manuscript fully available (please refer to the Data Availability Statement at the start of the manuscript PDF file)?

Reviewer #1: Yes

Reviewer #3: No

5. Is the manuscript presented in an intelligible fashion and written in standard English?

Reviewer #1: Yes

Reviewer #3: Yes

6. Review Comments to the Author

Reviewer #1: The authors have addressed the comments made in my first revision of their article. The paper reads more clearly and each of the vaguer issues pointed out have been elaborated on adequately.

Reviewer #3: (No Response)

7. PLOS authors have the option to publish the peer review history of their article (what does this mean?). If published, this will include your full peer review and any attached files.

**Do you want your identity to be public for this peer review?** For information about this choice, including consent withdrawal, please see our Privacy Policy.

Reviewer #1: No

Reviewer #3: No
